# ROGA: RANDOM OVER-SAMPLING BASED ON GENETIC ALGORITHM

## ABSTRACT

When using machine learning to solve practical tasks, we often face the problem of class imbalance. Unbalanced classes will cause the model to generate preferences during the learning process, thereby ignoring classes with fewer samples. The oversampling algorithm achieves the purpose of balancing the difference in quantity by generating a minority of samples. The quality of the artificial samples determines the impact of the oversampling algorithm on model training. Therefore, a challenge of the oversampling algorithm is how to find a suitable sample generation space. However, too strong conditional constraints can make the generated samples as non-noise points as possible, but at the same time they also limit the search space of the generated samples, which is not conducive to the discovery of better-quality new samples. Therefore, based on this problem, we propose an oversampling algorithm ROGA based on genetic algorithm. Based on random sampling, new samples are gradually generated and the samples that may become noise are filtered out. ROGA can ensure that the sample generation space is as wide as possible, and it can also reduce the noise samples generated. By verifying on multiple datasets, ROGA can achieve a good result.

## 1 INTRODUCTION

When modeling the classification problem, the balance of classes can ensure that the information is balanced during the learning process of the model, but the classes are often unbalanced in actual tasks(Kotsiantis et al. (2005); He & Garcia (2009)), which leads to the machine learning model preferring majority class samples. In order to solve this problem, a commonly used method uses oversampling algorithm to increase the minority class samples to balance the gap between the minority class samples and the majority class samples. Therefore, the quality of the generated samples will determine the training quality of the model after oversampling, but it is difficult to characterize this effect. In the past studies, oversampling methods only can estimated the noise samples, such as overlapping samples or outliers, and eliminated it as much as possible. Ensure that the sample does not make training the model more difficult.

SMOTE(Chawla et al. (2002)) and its derivative algorithms generate new samples by interpolation. There has been a lot of work in the selection of samples to be interpolated, the number of interpolations, or the method of interpolation, and the final goal is to reduce the generation of noise samples by restricting the interpolation position. Some researchers pointed out that sampling should not be performed in the sample space, but should be projected to other spaces(Wang et al. (2007), Zhang & Yang (2018)). In addition, some researchers have proposed that sampling should be taken from the minority distribution (Bellinger et al. (2015), Das et al. (2015)) to ensure the consistency of the distribution. It can be found that in order to reduce the generation of noise samples, the space for generating samples needs to be restricted, that is, the oversampling algorithm can only sample within a limited range.

However, just paying attention to noise generation may not achieve better results. Too strong constraints will cause the sample generation space to be limited, and the generated samples will fall into a local optimal solution, so that it is impossible to find samples that are more conducive to model training. Therefore, in order to reduce the limitation on the sample generation space and reduce noise as much as possible, this paper proposes oversampling algorithm ROGA based on genetic algorithm. ROGA will randomly sample in the feature space before the first iteration, and generate a

set of artificial samples as the basic population for the next iteration of the genetic algorithm. Random sampling will increase the noise in the samples, but it will also prevent ROGA from falling into a local optimal solution. In order to judge the noise, ROGA will calculate the fitness of each sample based on the Gaussian similarity with surrounding neighbors. The fitness represents the possibility that the sample is a noise sample. In the iterative process, new samples are generated through crossover and mutation operations, and samples with lower fitness are gradually removed, so as to reduce noise.

ROGA will generate new samples in the entire feature space, mainly because the initial population is generated by random sampling in the entire feature space. The crossover and mutation operations of genetic algorithms will continuously change the samples in the population to find samples with higher fitness values, and the screening mechanism will help us eliminate noisy samples in the population. Therefore, ROGA can balance the wide generation space and noise generation. Through experiments on multiple datasets, we found that ROGA can achieve the best scores on some datasets on the F1-score.

## 2 RELATED WORK

### 2.1 CLASS IMBALANCE

Data-based methods are represented by oversampling and undersampling. In order to balance the gap in the number of samples, oversampling generates new minority class samples, while undersampling(Arefeen et al. (2020); Koziarski (2020)) reduces redundant majority class samples.

The algorithm-based method is to improve the model algorithm to make it more suitable for training on unbalanced data. The first improvement measure is to balance the loss caused by the majority class and the minority class in the cost-sensitive loss function. Cost-sensitive can be combined with various machine learning algorithms, such as Adaboost(Wei (1999)), SVM(Lin (2002)), Decision Tree(Ling et al. (2005)), cross entropy(Li & Zhu (2019)) and so on. The second improvement measure is integrated learning technology. Liu et.al(Liu et al. (2009)) proposed EasyEnsemble and BalanceCascade, performed multiple under-sampling and integrated multiple models. Some researchers have added the ideas of Bagging(Wang & Yao (2009)) and Boosting(Chawla et al. (2003)) on SMOTE.

### 2.2 OVERSAMPLING

The oversampling algorithm is to generate new minority class samples to balance the quantitative difference between classes. The current oversampling algorithm can be divided into random sampling method, interpolation method, distributed sampling method and copy replacement method.

The random sampling method is to perform random sampling in the feature space, and the generated samples can easily deviate from the distribution of the original samples and produce noisy data. The interpolation oversampling algorithm represented by SMOTEChawla et al. (2002) restricts the generation of samples between two samples, and generates new minority class samples through linear interpolation. On the basis of the SMOTEChawla et al. (2002) algorithm, a large number of improved algorithms have appeared.

First, in order to further reduce the generation of noise, improvements can be made in the selection of the sample to be interpolated, the interpolation weight of the sample, and the position of the interpolation. SMOTE-Borderline(Nguyen et al. (2011)) believes that the minority class samples located at the decision boundary should be selected, and some researchers believe that interpolation should be performed within the minority class sample clusters generated by the clustering algorithm, such as KMeans-SMOTE(Douzas & Bacao (2018)), CURE-SMOTE(Ma & Fan (2017)), DBSMOTE(Bunkhumpornpat et al. (2012)), SOMO(Douzas & Bacao (2017b)), etc. ADASYN(He et al. (2008)), SMOTE-D(Torres et al. (2016)) and MSMOTE(Hu et al. (2009)) reduce the number of interpolations for some samples by setting the weight of the sample. When Safe-level SMOTE(Bunkhumpornpat et al. (2009)) generates samples, it will be closer to more regions of similar samples. In addition to controlling the noise generation in the generation stage, you can also filter the noises in the samples after generating the samples, such as SMOTE-FRST(Verbiest et al. (2012)), SMOTE-IPF(José et al. (2015)), MDO( Abdi & Hashemi (2015)), etc.

Second, the above methods all use linear interpolation when generating samples, but linear interpolation has limitations. Therefore, an improved algorithm for generating samples by nonlinear interpolation has appeared. Random-SMOTE(Dong & Wang (2011)) performs interpolation in a triangular area formed by three samples, and Geometric SMOTE(Douzas & Bacao (2017a)) generates samples in a geometric area instead of a linear area.

Third, choose a more appropriate generation space, such as projecting to low-dimensional space(Wang et al. (2007)) or high-dimensional space(Zhang & Yang (2018)). Fourth, when selecting neighbor samples, other indicators can be used to measure the distance between samples, such as cosine distance(Koto (2014)), propensity score(Rivera et al. (2014)), surrounding-neighbors(Garcia et al. (2012)) etc.

The distribution sampling method is to learn the distribution function or density function of minority class samples, and then sample from the distribution function to obtain a new artificial sample. DEAGO(Bellinger et al. (2015)) generates samples by denoising autoencoder, while RACOG and WRACOG(Das et al. (2015)) used the dependency tree algorithm to estimate the discrete probability distribution, and then generated artificial samples from the probability distribution through Gibbs sampling. The copy replacement method is to copy safe minority class samples and eliminate unsafe majority class samples(Stefanowski & Wilk (2008), Napierala et al. (2010)).

## 2.3 GENTIC ALGORITHM

Genetic algorithm (GA) is a bionic algorithm proposed by Holland(Holland (1975)) in 1975. Its main idea is to simulate the inheritance and mutation of genes during human evolution, and to control the evolution direction of samples through the fitness function. As a heuristic algorithm, it is often used in optimization problems(Rahmatsamii & Michielssen (1999), Fonseca & Fleming (1993), Wright (1990), Homaifar et al. (1994)), using the entire problem space as the search space, and searching for the best value in the search space. Genetic algorithms are also often used to optimize machine learning algorithms, such as SVM(Kim et al. (2005)), K-means(Krishna & Narasimha Murty (1999)), Decision Tree(Bala et al. (1995)), neural network(Montana (1989)), etc.

Genetic algorithm relies on randomness in the optimization process, which means that the speed of finding the optimal solution is unstable, but the genetic algorithm will not get stuck in a local solution when the fitness function is appropriate. In the process of solving, because it does not rely on the additional knowledge, its search range is wide. It is for this reason that we believe that genetic algorithms can find great patterns that have not yet been discovered.

## 3 ROGA

### 3.1 GA

#### 3.1.1 BASIC POPULATION

In the first iteration, this paper randomly generated $N$ artificial samples as the basic population. In each subsequent iteration, the artificial samples generated in the previous iteration will be used as the basic population for this iteration.

#### 3.1.2 FITNESS

The consideration of the degree of adaptation of the sample is mainly based on its attribution as a minority class sample. The fitness value of this sample is calculated by the Gaussian similarity with the surrounding samples, which is computed as:

$$fitness\left(x_i\right) = \sum_j^K y_j * sim\left(x_i, x_j\right) \tag{1}$$

$$sim\left(x_i, x_j\right) = e^{-\frac{\left(\sqrt{\Sigma_k^n \left(x_{ik} - x_{jk}\right)^2}\right)^2}{2\sigma^2}} \tag{2}$$

$x_i$ is the sample that currently needs to calculate fitness. $K$ is the number of neighbors to choose and set to 5 in this paper. $y_j \in \{-1, 1\}$. If $x_i$ and $x_j$ belong to the same sample, then $y_j = 1$. If they belong to the heterogeneous sample, then $y_j = -1$. $\sigma$ is the standard deviation of Gaussian kernel and set to 1 in this paper.

### 3.1.3 SELECTION

Before each crossover and mutation operation, $M$ candidates need to be selected from the population based on fitness. The candidates for crossover operation and mutation operation are $M$ samples with the lowest fitness in the population, and $M$ must be an even number. In the traditional genetic algorithm, the crossover operation should select the sample with the highest fitness in the population, hoping that the excellent genes can be inherited to the offspring. However, in the generation of minority samples, each feature value of the sample cannot establish a mapping relationship with the quality of the sample. Therefore, it is impossible to achieve the purpose of inheriting excellent genes when performing crossover operations. On the contrary, it will degrade the samples with high fitness in the population. Therefore, this paper only selects samples with low fitness to perform crossover and mutation operations and retains samples with high fitness in the current population.

### 3.1.4 CROSSOVER

Each crossover operation will generate two new samples from two candidate samples. For each pair of candidate samples$(x_i, x_j)$, each sample contains $m$ feature values $x_i = \{f_{i1}, f_{i2}, \ldots, f_{im}\}$, and $p$ feature values are randomly selected. For any pair of features $f_{ik}$ and $f_{jk}$, new sample is generated for linear interpolation, as shown in Eq. 3 and Eq. 4.

$$f'_{ik} = \alpha * f_{ik} + (1 - \alpha) * f_{jk} \tag{3}$$
$$f'_{jk} = \alpha * f_{jk} + (1 - \alpha) * f_{ik} \tag{4}$$

$\alpha \in [0, 1]$ is randomly generated.

### 3.1.5 MUTATION

Each mutation operation will get a new sample. For each candidate sample $x_i = \{f_{i1}, f_{i2}, \ldots, f_{im}\}$, randomly select $p$ feature values in the candidate sample, and multiply the selected feature value with randomly generated $\beta \in [0, 2]$ as the feature value of the new sample at that position, as shown in Eq. 5. The unselected feature value directly inherits the feature value of the original sample at that position.

$$f'_{ik} = \beta * f_{ik} \tag{5}$$

### 3.1.6 UPDATE

After the crossover and mutation operations, $3M$ new samples will be obtained. After calculating the fitness of all samples, the $N$ highest-fitting samples are selected as the basic population for the next iteration.

## 3.2 ROGA

The ROGA algorithm is divided into two stages, a random sampling stage and a genetic algorithm stage. The pseudo code is shown in Algorithm 1. In the random sampling stage, random sampling is performed in the feature space to obtain the required initial population $P$. At this stage, the number of samples in the population needs to be calculated first. In this paper, the number of basic populations $N_{syn}$ is set to the difference in quantity between majority class samples and minority class samples. The method of random sampling is that randomly samples a value within its range of values for each feature $f \in \{f_1, f_2, f_3, \ldots, f_n\}$. In the genetic algorithm stage, new artificial samples are generated in each iteration based on the obtained basic population and the artificial samples with low fitness values are eliminated. First, you need to determine the candidate samples for crossover and mutation operations. Calculate the fitness of each sample in the current population

Table 1: Dataset Description

| Name | Ratio | Number | Feature Number |
|------|-------|--------|----------------|
| Ecoil | 8.6:1 | 336 | 7 |
| optical_digits | 9.1:1 | 5620 | 64 |
| Satimage | 9.3:1 | 6435 | 36 |
| pen_digits | 9.4:1 | 10992 | 16 |
| sick_euthyroid | 9.8:1 | 3163 | 42 |
| spectrometer | 11:1 | 531 | 93 |
| car_eval_34 | 12:1 | 1728 | 21 |
| isolet | 12:1 | 7797 | 617 |
| us_crime | 12:1 | 1994 | 100 |
| thyroid_sick | 15:1 | 3772 | 52 |
| solar_flare_m0 | 19:1 | 1389 | 32 |
| oil | 22:1 | 937 | 49 |
| car_eval_4 | 26:1 | 1728 | 21 |
| wine_quality | 26:1 | 4898 | 11 |
| letter_img | 26:1 | 20000 | 16 |
| yeast_me2 | 28:1 | 1484 | 8 |
| mammography | 42:1 | 11183 | 6 |

in the sample space through Eq. 1 and Eq. 2, and then select the $M$ samples with the lowest fitness. $M$ is set to $N_{syn}/2$ in this article. After performing crossover and mutation operations on candidate samples, $3M$ new samples $P'$ will be obtained, and $N_{syn}$ samples with the highest fitness will be selected from $P \bigcup P'$ as the basic population for the next iteration.

---

**Algorithm 1** ROGA

---

**Require:** $X$:all of original samples; $Y$:all of labels; $K$:number of neighbors; $\alpha$: Gaussian kernel variance; $T$:iteration number;
**Ensure:** $P$: synthetic samples
1: $N_{syn}$ =Number of synthetic samples
2: $P = RandomSampling(N_{syn})$
3: **for** $i \in [1, T]$ **do**
4:   **for** $x \in P$ **do**
5:     Find $K$ nearest neighbors, and compute fitness by Eq. 1 and Eq. 2
6:   **end for**
7:   $Candidates$=select $M$ samples with the lowest fitness from $P$
8:   $SynSamples_{cross} = CrossoverOperation(Candidates)$
9:   $SynSamples_{mutation} = Mutation(Candidates)$
10:   $P' = SynSamples_{cross} \bigcup SynSamples_{mutation}$
11:   $P = Update(P, P')$
12: **end for**

---

## 4 EXPERIMENT

### 4.1 DATASET

The datasets come from UCI database(Dua & Graff (2017)), and the detailed informations of the datasets are shown in Table 1. The classification goals of all datasets are binary classification problems. If the goal of the dataset is a multi-classification problem, a certain class will be set as the target class, and the remaining categories will be divided into non-target classes and converted into a binary classification problem. Each dataset will be divided into 80% training set and 20% test set.

## 4.2 EVOLUTION

For the binary classification problem, a confusion matrix can be obtained on the results, as shown in Table 2. TP represents the number of positive classes predicted in the positive class, FP represents the number of predicted positive classes in the negative class, FN represents the number of negative classes predicted in the positive class, and TN represents the number of negative class predicted to be negative class. The confusion matrix can be used to calculate the accuracy, precision, recall and F1-score.

Table 2: Confusion Matrix

|          | Predicted Positive | Predicted Negative |
|----------|--------------------|--------------------|
| Positive | TP                 | FN                 |
| Negative | FP                 | TN                 |

Accuracy can be defined as:

$$\text{Accuracy} = \frac{TP + TN}{TP + TN + FP + FN} \tag{6}$$

Precision can be defined as:

$$\text{Precision} = \frac{TP}{TP + FP} \tag{7}$$

Recall can be defined as:

$$\text{Recall} = \frac{TP}{TP + FN} \tag{8}$$

F1-score can be defined as:

$$F1 = 2 \cdot \frac{\text{Recall} \cdot \text{Precision}}{\text{Recall} + \text{Precision}} \tag{9}$$

## 4.3 PERFORMANCE

This paper uses six oversampling algorithms to compare with ROGA, including Baseline, SMOTE(Chawla et al. (2002)), SMOTE-Borderline1(Nguyen et al. (2011)), ADASYM(He et al. (2008)), KMeans-SMOTE(Douzas & Bacao (2018)), and SMOTENC(Chawla et al. (2002)). Baseline represents the unused oversampling algorithm. The experimental results are shown in Table 3. Due to the large fluctuations in the experimental results of ROGA, the experimental results of ROGA are the best results among the 20 experiments. The evaluation method used in the table is F1-score, and the classification algorithm uses XGBoost(Chen & Guestrin (2016)). For each experiment, the hyperparameters of classification algorithm are fixed. The remaining metrics will be presented in the appendix. † represents the best score in this dataset.

From Table 3, it can be found that ROGA has achieved the best results on most datasets, and even the improvement effect is more obvious, such as ecoil. ROGA and other oversampling algorithms both measure noise on the distribution to remove noise, and ROGA with a wider range generates samples that are more conducive to model training.

At the same time, it is also found that artificial samples generated by the oversampling algorithm on some datasets lead to lower scores on the test set. This shows that the generation space constrained by the oversampling algorithm is not suitable for the distribution of all datasets, which can only balance the number of categories and cannot provide benefits to the training of the model. The limited generation range also causes the oversampling algorithm to be unable to deviate from the current local optimal solution. In order to avoid this kind of solidification, ROGA will perform random sampling in the entire sample space before first iteration. This random sampling can make ROGA deviate from the current local optimum and achieve better results. From the results on the isolet dataset, it can be seen that the oversampling algorithms as a comparison are lower than the baseline in the test set score, while ROGA improves the effect of the model.

Table 3: Performace

| Name | Base | SMOTE | Borderline1 | Adasyn | Kmeans | NC | ROGA |
|------|------|-------|-------------|--------|--------|-----|------|
| Ecoil | 0.7873 | 0.8541 | 0.8205 | 0.7662 | 0.7873 | 0.7873 | 0.9413$^\dagger$ |
| optical_digits | 0.9688 | 0.9628 | 0.9633 | 0.9718 | 0.9641 | 0.9669 | 0.9737$^\dagger$ |
| Satimage | 0.8542$^\dagger$ | 0.8311 | 0.8379 | 0.8352 | 0.8503 | 0.8405 | 0.8455 |
| pen_digits | 0.9855 | 0.9886 | 0.9897 | 0.9876 | 0.9887 | 0.9855 | 0.9897$^\dagger$ |
| sick_euthyroid | 0.9462 | 0.9413 | 0.9493 | 0.9539$^\dagger$ | 0.9462 | 0.9377 | 0.9493 |
| spectrometer | 0.8626 | 0.8969 | 0.8334 | 0.8969 | 0.8626 | 0.8721 | 0.9382$^\dagger$ |
| car_eval_34 | 0.9616$^\dagger$ | 0.9057 | 0.8922 | 0.9000 | 0.9279 | 0.9057 | 0.9423 |
| isolet | 0.9158 | 0.9069 | 0.8952 | 0.9089 | 0.9158 | 0.9069 | 0.9266$^\dagger$ |
| us_crime | 0.7017 | 0.7035 | 0.6848 | 0.7028 | 0.6909 | 0.6880 | 0.7448$^\dagger$ |
| thyroid_sick | 0.9502 | 0.9332 | 0.9435 | 0.9300 | 0.9354 | 0.9300 | 0.9627$^\dagger$ |
| solar_flare_m0 | 0.4867 | 0.5184 | 0.5184 | 0.5123 | 0.4844 | 0.5206 | 0.5728$^\dagger$ |
| oil | 0.7254 | 0.7258 | 0.7112 | 0.7616 | 0.7112 | 0.6867 | 0.7864$^\dagger$ |
| car_eval_4 | 0.9616 | 0.9249 | 0.9329 | 0.9499 | 0.9576 | 0.9616$^\dagger$ | 0.9576 |
| wine_quality | 0.5731 | 0.6856 | 0.6655 | 0.6918 | 0.6768 | 0.7001$^\dagger$ | 0.6275 |
| letter_img | 0.9815$^\dagger$ | 0.9746 | 0.9792 | 0.9747 | 0.9814 | 0.9746 | 0.9771 |
| yeast_me2 | 0.7291 | 0.6895 | 0.7431 | 0.7416 | 0.6806 | 0.7416 | 0.7445$^\dagger$ |
| mammography | 0.8759 | 0.8175 | 0.8334 | 0.8095 | 0.8871 | 0.8643 | 0.9015$^\dagger$ |

## 4.4 LIMITATION

ROGA generates the basic population through random sampling, which expands the sample generation space on the one hand, but also increases uncertainty. Therefore, each artificial sample generated by ROGA is not fixed, and compared with other oversampling algorithms, this uncertainty is more serious.

Figure 1 shows that the ROGA model test results are unstable. The uncertainty contained in the initial population generation will make the model after training better, or it may not improve the model effect significantly. Therefore, when using ROGA to solve class imbalance, it is necessary to conduct as many experiments as possible to find the artificial samples that perform best on a specific metric. However, under the premise that the influence of artificial samples cannot be well described, the traditional oversampling paradigm that try to compensate for the impact of class imbalance through an experiment is unrealistic. Multiple sampling and evaluation are more conducive to taking advantage of oversampling.

## 5 CONCLUSION

In order to avoid the generation of noise, the current oversampling algorithm performs sampling in a limited generation space. However, the limited generation space will cause the oversampling algorithm to fall into a local optimal solution, so that it cannot effectively generate artificial samples that are beneficial to model learning. In order to balance the extensive generation space and noise generation, this paper proposes the ROGA algorithm. Use random sampling to generate the initial sample population to ensure sample generation space. Then, new artificial samples are continuously generated through genetic algorithm, and noise points in the population are eliminated according to fitness. Experiments have proved that ROGA has achieved the best F1-score on multiple datasets. Therefore, the wide generation space is conducive to the oversampling algorithm to collect more high-quality samples, which can not only balance the difference in quantity but also benefit the model learning.

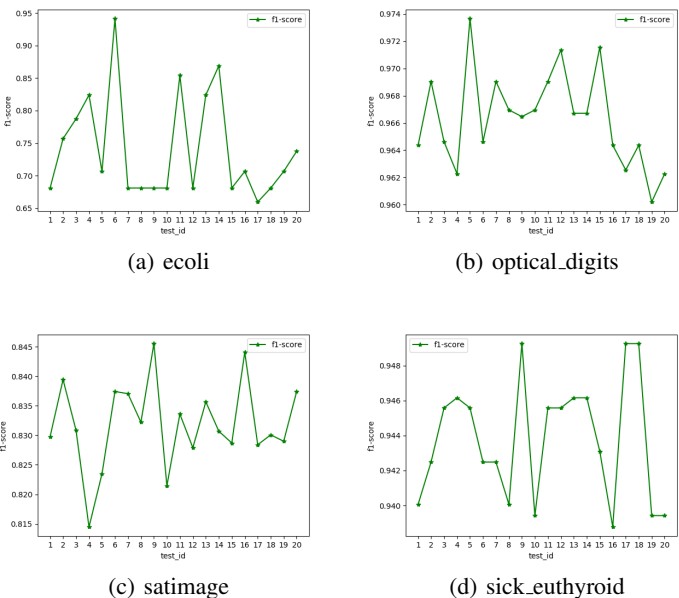

(a) ecoli

(b) optical_digits

(c) satimage

(d) sick_euthyroid

Figure 1: F1-score in 20 experiments. This paper chooses four datasets to preform the uncertainty of ROGA. $test\_id$ represents the experiment index and vertical axis represents the F1-score. The value range of the vertical axis of each graph is not the same. Although the amplitude is similar, it does not mean that the range of change is the same.

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

## A    APPENDIX

Table 4: Precision

| Name | Base | SMOTE | Borderline1 | Adasyn | Kmeans | NC | ROGA |
|------|------|-------|-------------|--------|--------|----|----|
| ecoil | 0.7873 | 0.8269 | 0.7792 | 0.7156 | 0.7873 | 0.7873 | $0.9938^{\dagger}$ |
| optical_digits | $0.9869^{\dagger}$ | 0.9713 | 0.9650 | 0.9769 | 0.9859 | 0.9792 | 0.9842 |
| Satimage | $0.8866^{\dagger}$ | 0.8086 | 0.8226 | 0.8112 | 0.8824 | 0.8207 | 0.8802 |
| pen_digits | 0.9955 | $0.9961^{\dagger}$ | 0.9963 | 0.9942 | 0.9944 | 0.9955 | 0.9946 |
| sick_euthyroid | 0.9717 | 0.9506 | 0.9779 | 0.9686 | 0.9717 | 0.9494 | $0.9779^{\dagger}$ |
| spectrometer | 0.9758 | 0.9418 | 0.9720 | 0.9418 | 0.9758 | 0.9341 | $0.9876^{\dagger}$ |
| car_eval_34 | 0.8812 | 0.8604 | 0.8485 | 0.8529 | 0.8999 | 0.8604 | $0.9134^{\dagger}$ |
| Isolet | 0.9608 | 0.9043 | 0.9018 | 0.9142 | $0.9608^{\dagger}$ | 0.9043 | 0.9532 |
| us_crime | 0.8256 | 0.6898 | 0.6819 | 0.6996 | 0.7793 | 0.6679 | $0.8743^{\dagger}$ |
| thyroid_sick | $0.9739^{\dagger}$ | 0.9217 | 0.9317 | 0.9085 | 0.9440 | 0.9085 | 0.9672 |
| solar_flare_m0 | 0.4755 | 0.5263 | 0.5263 | 0.5146 | 0.4753 | 0.5320 | $0.6447^{\dagger}$ |
| oil | 0.7390 | 0.6994 | 0.7112 | 0.7240 | 0.7112 | 0.6707 | $0.8037^{\dagger}$ |
| car_eval_4 | 0.9512 | 0.9067 | 0.9337 | 0.9306 | 0.9964 | 0.9512 | $0.9964^{\dagger}$ |
| wine_quality | 0.6856 | 0.6581 | 0.6471 | 0.6613 | 0.6471 | 0.6778 | $0.7785^{\dagger}$ |
| letter_img | 0.9958 | 0.9733 | 0.9765 | 0.9709 | $0.9988^{\dagger}$ | 0.9733 | 0.9868 |
| yeast_me2 | 0.7431 | 0.6506 | 0.7217 | 0.6958 | 0.7060 | 0.6958 | $0.7788^{\dagger}$ |
| mammography | 0.9337 | 0.7693 | 0.7917 | 0.7529 | 0.9365 | 0.8400 | $0.9479^{\dagger}$ |

Table 5: Recall

| Name | Base | SMOTE | Borderline1 | Adasyn | Kmeans | NC | ROGA |
|------|------|-------|-------------|--------|--------|----|----|
| Ecoil | 0.7873 | 0.8873 | 0.8810 | 0.8684 | 0.7873 | 0.7873 | $0.9000^{\dagger}$ |
| optical_digits | 0.9523 | 0.9546 | 0.9617 | 0.9668 | 0.9445 | 0.9555 | $0.9637^{\dagger}$ |
| Satimage | 0.8279 | 0.8587 | 0.8554 | $0.8651^{\dagger}$ | 0.8244 | 0.8640 | 0.818 |
| pen_digits | 0.9760 | 0.9815 | 0.9833 | 0.9813 | 0.9831 | 0.9760 | $0.9849^{\dagger}$ |
| sick_euthyroid | 0.9240 | 0.9325 | 0.9247 | $0.9403^{\dagger}$ | 0.9240 | 0.9268 | 0.9247 |
| spectrometer | 0.8000 | 0.8624 | 0.7667 | 0.8624 | 0.8000 | 0.8291 | $0.9000^{\dagger}$ |
| car_eval_34 | 0.9588 | 0.9693 | 0.9538 | 0.9681 | 0.9614 | 0.9693 | $0.9769^{\dagger}$ |
| isolet | 0.8800 | 0.9095 | 0.8889 | 0.9036 | 0.8800 | $0.9095^{\dagger}$ | 0.9035 |
| us_crime | 0.6509 | $0.7201^{\dagger}$ | 0.6877 | 0.7061 | 0.6488 | 0.7158 | 0.6865 |
| thyroid_sick | 0.9289 | 0.9455 | 0.9561 | 0.9544 | 0.9272 | 0.9544 | $0.9583^{\dagger}$ |
| solar_flare_m0 | 0.4985 | 0.5158 | 0.5158 | 0.5113 | 0.4940 | 0.5173 | $0.5528^{\dagger}$ |
| oil | 0.7134 | 0.7623 | 0.7112 | 0.8178 | 0.7112 | 0.7067 | $0.7711^{\dagger}$ |
| car_eval_4 | 0.9725 | 0.9451 | 0.9226 | 0.9714 | 0.9250 | $0.9726^{\dagger}$ | 0.9250 |
| wine_quality | 0.5488 | 0.7276 | 0.6904 | $0.7399^{\dagger}$ | 0.7259 | 0.7301 | 0.5876 |
| letter_img | 0.9681 | 0.9759 | $0.9818^{\dagger}$ | 0.9787 | 0.9653 | 0.9759 | 0.9678 |
| yeast_me2 | 0.7167 | 0.7626 | 0.7695 | 0.8209 | 0.6611 | $0.8209^{\dagger}$ | 0.7181 |
| mammography | 0.8322 | 0.8891 | 0.8904 | $0.9036^{\dagger}$ | 0.8481 | 0.8926 | 0.8642 |

Table 6: Accuracy

| Name | Base | SMOTE | Borderline1 | Adasyn | Kmeans | NC | ROGA |
|------|------|-------|-------------|--------|--------|-----|------|
| Ecoil | 0.9524 | 0.9643 | 0.9524 | 0.9286 | 0.9524 | 0.9524 | 0.9881$^\dagger$ |
| optical_digits | 0.9900 | 0.9879 | 0.9879 | 0.9907 | 0.9886 | 0.9893 | 0.9915$^\dagger$ |
| Satimage | 0.9528 | 0.9360 | 0.9403 | 0.9372 | 0.9512$^\dagger$ | 0.9403 | 0.9503 |
| pen_digits | 0.9949 | 0.9960 | 0.9964 | 0.9956 | 0.9960 | 0.9949 | 0.9964$^\dagger$ |
| sick_euthyroid | 0.9798 | 0.9772 | 0.9810 | 0.9823$^\dagger$ | 0.9798 | 0.9760 | 0.9810 |
| spectrometer | 0.9549 | 0.9624 | 0.9474 | 0.9624 | 0.9549 | 0.9549 | 0.9774$^\dagger$ |
| car_eval_34 | 0.9154 | 0.9676 | 0.9630 | 0.9653 | 0.9769 | 0.9676 | 0.9815$^\dagger$ |
| isolet | 0.9790 | 0.9744 | 0.9718 | 0.9754 | 0.9790 | 0.9744 | 0.9810$^\dagger$ |
| us_crime | 0.9519 | 0.9299 | 0.9299 | 0.9339 | 0.9479 | 0.9218 | 0.9579$^\dagger$ |
| thyroid_sick | 0.9905 | 0.9862 | 0.9883 | 0.9852 | 0.9873 | 0.9852 | 0.9926$^\dagger$ |
| solar_flare_m0 | 0.9482$^\dagger$ | 0.9282 | 0.9282 | 0.9195 | 0.9396 | 0.9310 | 0.9454 |
| oil | 0.7617 | 0.9532 | 0.9574 | 0.9574 | 0.9574 | 0.9489 | 0.9702$^\dagger$ |
| car_eval_4 | 0.9931 | 0.9861 | 0.9884 | 0.9907 | 0.9931 | 0.9931 | 0.9931$^\dagger$ |
| wine_quality | 0.9665 | 0.9527 | 0.9527 | 0.9527 | 0.9494 | 0.9576 | 0.9698$^\dagger$ |
| letter_img | 0.9976 | 0.9966 | 0.9972 | 0.9966 | 0.9976$^\dagger$ | 0.9966 | 0.9970 |
| yeast_me2 | 0.9757 | 0.9596 | 0.9730 | 0.9677 | 0.9730 | 0.9677 | 0.9784$^\dagger$ |
| mammography | 0.9903 | 0.9803 | 0.9828 | 0.9782 | 0.9911 | 0.9871 | 0.9921$^\dagger$ |

