# OpenReview forum: "ROGA: Random Over-sampling Based on Genetic Algorithm"
_ICLR.cc/2021/Conference — Reject_

### Official Review · AnonReviewer3 · 2020-10-27
**Review ROGA**

**Rating:** 3
**Confidence:** 5

**Review:**

The authors tackle the problem of class imbalance in supervised learning. They propose an oversampling algorithm that is based on a genetic algorithm. Samples are iteratively generated and filtered to maintain informative samples. The authors demonstrate the efficacy of the approach using several examples.

Although the problem is important and the solution might be interesting, I feel that the paper is not scientifically sound and the English level is not satisfactory. Below I will detail some of the problems with the current version of the manuscript:

-First paragraph: “only can estimated” - this is is not in English.

-First paragraph: “Ensure that the sample does not make…” - again this sentence does not make sense.

-Authors did not use the correct citation format, please see instructions by ICLR.

-P2: SMOTEChawla - here the citation is mixed with the method name.

-P3: Holland - the year is mentioned twice.

-2.3 has the title “Genetic Algorithm” and 3.1 has the title “GA”, aren’t these the same???

-3.1 the paper does not generate anything, perhaps you meant the method?

-Equation (2)  x_ik is not defined, what are the two indexes for? Sample and feature? Need to define.

-Equation (2) you take the square of the square root, these cancel each other!

-The authors mostly cite SMOTE based method for class imbalance, but there are other methods such as:

[1] Lindenbaum, Ofir, Jay Stanley, Guy Wolf, and Smita Krishnaswamy. "Geometry based data generation." In Advances in Neural Information Processing Systems, pp. 1400-1411. 2018.

-Perhaps you could use bar plots for figure 1 instead of the scatter plots which are not that informative.

There are many other mistakes, so I feel that at this stage the paper can not be published at this venue.

---

### Official Review · AnonReviewer2 · 2020-10-28
**Official Blind Review #2**

**Rating:** 5
**Confidence:** 4

**Review:**

The authors focus on the class imbalance problem and propose an algorithm named ROGA to generated samples of minority classes to balance the quantitative difference between classes. Specifically, the proposed ROGA generates samples of minority classes by using a Genetic Algorithm (GA) to explore the sample space. Moreover, to reduce the noise samples generated by ROGA, the authors propose to calculate the fitness as the Gaussian similarity with the surrounding samples and eliminate the samples with low fitness. My detailed comments are as follows.

Positive points:
1. The authors propose an algorithm named ROGA to generate samples of minority classes to balance the quantitative difference between classes.

2. To reduce the noise samples, the authors propose to calculate the fitness as the Gaussian similarity with the surrounding samples and eliminate the samples with low fitness.

3. Experimental results on multiple datasets demonstrate the effectiveness of the proposed method.

Negative points:
1. The authors argue that the generation space of the proposed method is widened mainly due to the initial random sampling over the entire sample space. However, during the optimization process of GA, the sample space may be rapidly narrowed down, which makes the generation space much smaller. How to demonstrate that ROGA can ensure the sample generation space is as wide as possible during the optimization? More description and experimental analysis are required.

2. The experiments are insufficient in Section 4.3. In Table 3, the compared methods are SMOTE [1] and its extensions, which are not up-to-date. It would be better to compare ROGA with more closely related and up-to-date methods such as OHEM [2], Focal Loss [3], and SMOTE-WENN[4].

3. It is not clear why the samples with low fitness are selected to perform the crossover and mutation operation. Does the algorithm converge effectively under this setting?

4. The reason behind the choice of the mutation operation is not clear. Why the feature value can only be positively scaled?

5. Lots of experimental details such as hyper-parameters $p$, $\beta$, training iterations, and the probability of crossover/mutation are missing, which makes the paper hard to be reproduced.

6. In Table 3, why the performance of the proposed method on the wine_quality data set is much poorer than others. More explanations are required.

7. The ablation study is insufficient. $K$ plays an important role in sample generation. More results on different values of $K$ are required.

Minor issues:
1. In Section 2.2, “RACOG and WRACOG (Das et al. (2015)) used the dependency tree algorithm …” should be “RACOG and WRACOG (Das et al. (2015)) use the dependency tree algorithm …”.

2. In the Introduction, “oversampling methods only can estimated …” should be “oversampling methods only can estimate …”.

3. In Section 4.4, “traditional oversampling paradigm that try to …” should be “traditional oversampling paradigm that tries to …”.

References:

[1] Nitesh V. Chawla, Kevin W. Bowyer, Lawrence O. Hall, and W. Philip Kegelmeyer. Smote: Synthetic minority over-sampling technique. Journal of Artificial Intelligence Research, 16(1):321–357, 2002.

[2] Shrivastava, Abhinav & Mulam, Harikrishna & Girshick, Ross. (2016). Training Region-Based Object Detectors with Online Hard Example Mining. 761-769. 10.1109/CVPR.2016.89.

[3] Lin, Tsung-Yi & Goyal, Priyal & Girshick, Ross & He, Kaiming & Dollar, Piotr. (2018). Focal Loss for Dense Object Detection. IEEE Transactions on Pattern Analysis and Machine Intelligence. PP. 1-1. 10.1109/TPAMI.2018.2858826.

[4] Guan H, Zhang Y, Xian M, et al. SMOTE-WENN: Solving class imbalance and small sample problems by oversampling and distance scaling[J]. Applied Intelligence, 2020: 1-16.

---

### Official Review · AnonReviewer1 · 2020-10-28
**Methodological contribution lacks context and limited experimental design**

**Rating:** 3
**Confidence:** 4

**Review:**

In this work, the authors propose an oversampling technique which creates a population of synthetic samples for an imbalanced dataset using genetic algorithms. Each individual in the GA population corresponds to a synthetic sample, and the fitness function is based on the similarity (in feature space) of the synthetic sample compared to nearby samples of the same and different classes; standard crossover and mutation operators are used. In a limited set of experiments, the proposed approach sometimes outperforms competing approaches.

I think the basic idea of using a GA to avoid local minima in the feature space is reasonable, and the proposed fitness function makes sense, though it is perhaps a bit obvious. However, I believe the paper has several key limitations that need to be addressed.

The discussion on related work omits a large number of highly related references. In particular, a large number of existing papers propose to use genetic algorithms for oversampling, including: [Saladi and Dash, Soft Computing for Problem Solving, 2019], [Tallo and Musdholifah, ICST, 2018], [Ha and Lee, IMCOM 2016], and [Qiong et al., JDIM 2016]. While the proposed approach likely differs from these in some respects, the complete lack of context with respect to existing GA approaches for addressing class imbalance makes it difficult to judge the methodological novelty.

Second, the set of baselines considered in the work is rather limited. Thus, it is not clear if the paper meaningfully improves empirical performance compared to state of the art. In addition to lacking any other GA baselines, the experiments also lack a representative generative adversarial network-based approach. Considering the prevalence of such approaches, at least a standard implementation should be included in the comparison.

Third, the experimental design itself could be significantly improved. The current design does not allow for any estimate of variance in performance of the methods on each dataset. Thus, it is not possible to tell if any of the results are significant. Indeed, additional experiments suggest that the proposed approach is not stable, so that further calls into question the empirical advantage of this approach. A simple way to estimate such variances could be to perform cross-validation by bootstrapping the dataset (e.g., k-fold cross-validation). It is also not clear how the hyperparameters for the other baseline methods are selected.

It is also not clear how the approach may work for datasets with non-numeric features, such as text or images. Presumably, some standard embedding method (word2vec, CNNs, etc.) could be used to first embed the data to a vector space, but it is then not obvious to me how well the proposed GA approach would be in exploring that space.

Minor comments
---------------------

The paper has numerous small grammatical errors. I do not believe these affect understanding the paper, but it requires another round of proofreading.

There is some issue with the formatting of the references in the text; they are all followed by two closing parentheses.

For the crossover operation, is \alpha drawn uniformly from [0,1], or from a truncated Gaussian with a mean around 0.5? or something else?

In Algorithm 1, \alpha is used for the Gaussian kernel variance, while \sigma is used in the text.

For Figure 1, something like a box plot or violin plot is probably more appropriate for presenting that data.

The references are not consistently formatted.

---

### Official Review · AnonReviewer4 · 2020-11-02
**Interesting Idea - reservations about the results presented**

**Rating:** 4
**Confidence:** 3

**Review:**

The paper introduces ROGA - an oversampling method that uses a genetic algorithm to generate synthetic samples for the minority class. Results in classification scenarios with imbalance dataset demonstrate the effectiveness of the proposed method.

A lot of grammatical errors mar the readability of the manuscript. Resolving them through proofreading and copy-editing would improve the clarity of the paper.

The related work section is extensive and provides a good overview of related literature on methods that address class imbalances. However, the whole section reads like a laundry list of previous methods instead of painting a cohesive story about what has been done, what gaps remain and most importantly how the proposed method (ROGA) can address these gaps. It is difficult to place where ROGA fits in amongst previous literature.

Section 4.2 (definitions for accuracy, precision, recall, F1, TP, FP) is background and does not belong in the method section. I would suggest that these could be removed entirely from the paper as this are basic machine learning terms that are well known and understood.

The presented results seem to be the max score achieved across 20 independent experiments. Is this correct? If so, this will have high bias. Why not report the first order statistics based on the 20 experiments? For example, the mean score achieved, and standard deviation would enable the reader to have a much more balanced view of the method’s merits and limitation.
Further, it seems this max selection was not used for the other baselines in the comparative results. This would raise significant questions regarding the validity of the comparative results.

What is the added computational cost of ROGA versus a traditional oversampling method that does not use a population? Since the fitness uses K-nearest neighbor Gaussian similarity, the computation overhead can be substantial depending on how many iterations must be run. This should be reported as it is an important consideration in large scale deployments.

The novelty in the method is marginal. The principal contribution of the method is an attempt to leverage the diversity benefits of a population-based method (GA in this case) towards generating higher quality synthetic samples for oversampling. However, the results as they stand do not fully corroborate that this benefit translates effectively to improved performance.

---

### Decision · Program_Chairs · 2021-01-07
**Final Decision**

**Decision:**

Reject

**Comment:**

This paper proposes to address the class imbalance problem by defining an over-sampling strategy based on oversampling. It brings potentially interesting ideas. The reviewers agree on the fact that the experiments are limited, some methodological aspects require some clarifications and the writing needs to be improved.
The authors did not provide any rebuttal.
Hence I recommend rejection.